# The Molecular Interactions of ZIKV and DENV with the Type-I IFN Response

**DOI:** 10.3390/vaccines8030530

**Published:** 2020-09-14

**Authors:** Rosa C. Coldbeck-Shackley, Nicholas S. Eyre, Michael R. Beard

**Affiliations:** 1School of Biological Sciences, Research Centre for Infectious Diseases, The University of Adelaide, Adelaide, SA 5005, Australia; rosa.coldbeck-shackley@adelaide.edu.au; 2College of Medicine and Public Health, Flinders University, Bedford Park, SA 5042, Australia; nicholas.eyre@flinders.edu.au

**Keywords:** ZIKV, DENV, innate immunity, interferon, IFN, evasion

## Abstract

Zika Virus (ZIKV) and Dengue Virus (DENV) are related viruses of the *Flavivirus* genus that cause significant disease in humans. Existing control measures have been ineffective at curbing the increasing global incidence of infection for both viruses and they are therefore prime targets for new vaccination strategies. Type-I interferon (IFN) responses are important in clearing viral infection and for generating efficient adaptive immune responses towards infection and vaccination. However, ZIKV and DENV have evolved multiple molecular mechanisms to evade type-I IFN production. This review covers the molecular interactions, from detection to evasion, of these viruses with the type-I IFN response. Additionally, we discuss how this knowledge can be exploited to improve the design of new vaccine strategies.

## 1. Introduction

Dengue Virus (DENV) and Zika Virus (ZIKV) are closely related viruses of the *Flavivirus* genus and although most infections in humans are asymptomatic, both viruses can cause severe life threatening or debilitating disease. DENV includes four serotypes (DENV-1-4) and collectively causes the greatest disease burden of all flaviviruses, infecting more than 390 million people causing 21,000 deaths annually [1]. Comparatively, ZIKV is mono-serotypic and causes far fewer deaths, but infection during pregnancy can result in infection of the developing fetus, early pregnancy loss, or developmental and neurological impairment in newborns [2]. The primary mode of ZIKV and DENV transmission is via mosquitoes of the *Aedes* species. As a result of their shared vector, both viruses co-circulate in tropical and sub-tropical regions of Africa, Asia and the Americas [3]. It is estimated that more than one third of the world population lives in areas with high risk of DENV infection [4]. Worryingly, both of these viruses have rapidly expanded their geographic range in recent years as a result of factors such as climate change and globalization [5]. Currently, there are no specific antiviral treatments for either infection despite DENV and ZIKV posing serious threats to human health and placing large socio-economic burdens on many of the world’s most under-developed nations. To date there are no approved vaccines for ZIKV, but several vaccine candidates are currently being investigated and as of late 2019, the WHO listed 15 vaccine candidates in phase I/II clinical trials [6]. These include DNA, RNA, recombinant protein, recombinant viral vector, and inactivated whole virus vaccines. As for DENV, there is an approved vaccine (Dengvaxia^®^) developed by Sanofi Pasteur; however, it is not recommended for children under 9 years of age and has been associated with safety concerns due to its ability to increase the risk of severe disease in people who are seronegative when receiving vaccination [7]. Despite its limited applicability, Dengvaxia^®^ generates immune responses against all four serotypes and reduces the incidence of severe complications associated with DENV infection [8]. Development of specific, effective, and safe treatments and vaccines for the prevention of both ZIKV and DENV requires understanding of their fundamental biology, pathogenesis, and molecular interactions with the host immune response.

The type-I IFNs are produced in response to sensing viral infection within cells and act as a crucial front-line defense against a broad range of viral pathogens [9]. Once produced and secreted by infected cells these antiviral cytokines can act on the same cell or neighboring cells expressing their cognate receptor to initiate signaling that leads to the production of hundreds of interferon stimulated genes (ISGs) [10]. These ISGs carry out a range of direct antiviral, regulatory or immunomodulatory functions giving rise to an antiviral state within host cells and tissues [11]. Importantly this IFN response shapes aspects of the adaptive immune response to viruses, leading to improved cell mediated and humoral responses [12]. As a result, the IFN response is a barrier that viruses must overcome to cause infection, replicate, and spread. Not surprisingly, because of this strong selective pressure many viruses have evolved mechanisms to counteract IFN responses, allowing them to gain a foothold and cause infection. For example, flaviviruses such as ZIKV and DENV have evolved a complex array of molecular interactions with the host innate immune response to undermine both the production and downstream signaling of IFNs.

This review aims to summarize the current knowledge of the molecular interactions between ZIKV and DENV with the type-I IFN response. Furthermore, we discuss how this knowledge can be exploited in the development of safe and effective vaccines.

## 2. DENV/ZIKV Virology and Lifecycle

The design of new therapeutics and vaccines is underpinned by understanding the fundamental biology and lifecycle of the virus. In the following section we explain the relevant aspects of *Flavivirus* virology and lifecycle.

### 2.1. Phylogeny and Genome Structure

The *Flavivirus* are a genus of primarily arthropod borne, enveloped, non-segmented positive-sense single-stranded RNA (+ssRNA) viruses that include several major human pathogens such as Yellow Fever Virus (YFV), Japanese Encephalitis Virus (JEV), West Nile Virus (WNV), DENV and ZIKV. Within the *Flavivirus* genus DENV and ZIKV are closely related, sharing on average 55% amino acid identity [13]. Both the +ssRNA genomes for DENV and ZIKV are approximately 11kb in size, are flanked by 5′ and 3′ untranslated regions (UTRs) and contain a central open reading frame (ORF) encoding the structural (Capsid (C), pre-Membrane (prM) and Envelope (E)) and non-structural (NS1, NS2A, NS2B, NS3, NS4A, NS4B, NS5) viral proteins (Figure 1). The 5′ end of the genome terminates in a type-I cap, mimicking the structure of host mRNA [14]. Both the 5′ and 3′ UTR’s are predicted to form extensive RNA secondary structures that are conserved among flaviviruses [15]. These RNA structures are critical for genome replication and also evasion of host immune responses [16]. In general, the structural proteins are involved with infectious particle (virion) production, virus attachment, entry and uncoating. The NS proteins play various roles in the intracellular aspects of the virus lifecycle, including genome replication, translation, processing of the viral polyprotein, membrane rearrangements, virion assembly, and evasion of the innate immune response.

### 2.2. Lifecycle

As closely related flaviviruses, ZIKV and DENV share many aspects of their lifecycle within host cells. The *Flavivirus* lifecycle is carried out in several stages: Attachment, entry, fusion and uncoating, initial translation of the viral proteins, generation of the replication complex and replication of new viral genomes, assembly, maturation, and egress [18], summarized in Figure 2.

Initially virions attach to host cells via interactions with host cell surface receptors to gain entry to the cell. Virion attachment and entry involves multiple host surface receptors [19,20]. Some of the known attachment factors for DENV and ZIKV include glycosaminoglycans (GAGs), such as heparan sulphate [19], C-type Lectin Receptors (CLRs) such as DC-SIGN [21] and members of the TIM (TIM1, TIM3, and TIM4) and TAM (Tyro3, Axl, and Mer) family of receptors [21,22]. Following attachment, the virion enters the host cell via clathrin-mediated endocytosis [19]. The clathrin-coated vesicle is processed to form an early endosome, these are then increasingly acidified to form late-endosomes [23]. The low pH environment of the late-endosome drives conformational changes in the E protein present on the virion surface leading to membrane fusion, uncoating and release of the +ssRNA genome into the cytoplasm [18]. In the cytosol, viral replication is initiated by direct translation of the +ssRNA by the host ribosome machinery that recognizes the type-I cap in the 5′ UTR [18]. The ORF is translated as a single multi-pass transmembrane polyprotein imbedded into the endoplasmic reticulum (ER) membrane. Following and during translation, the polyprotein is cleaved by host and viral proteases (NS3) to liberate the individual viral proteins [15]. Together, multiple NS proteins interact with host factors to induce changes in the structure of the ER membranes to generate replication organelles or “vesicle packets” that house the viral replication complex (RC) [24]. The RC acts to concentrate viral and host proteins that are essential for viral genome replication and also functions to hide the replicating viral RNA from innate immune detection [24,25]. Following formation of the RC, the +ssRNA genome is copied through a -ssRNA intermediate by the NS5 RNA dependent RNA polymerase (RdRp) in coordination with other essential host and viral factors. The progeny genomes are capped by the NS5 methyl-transferase (MTase) domain and exit the vesicle packets through the vesicle pore where a single +ssRNA copy interacts with multiple copies of the C protein to form the nucleocapsid [24]. Next, this nucleoprotein complex buds into the ER-derived membrane imbedded with viral prM–E protein heterodimers that coat the newly enveloped immature virion [18,26]. After entering the ER lumen, the immature *Flavivirus* particle is shuttled through the secretory pathway. Here the virion undergoes maturation involving the acid-induced rearrangement of E protein and subsequent cleavage of the prM protein by furin into the mature M protein, forming a smooth outer coat [27]. Interestingly, some immature prM also remains on the surface of virions [28] and during DENV infection, heterotypic cross-reactive antibodies raised against prM promote antibody-dependent enhancement of severe dengue disease [29] as reviewed in [30]. After maturation, new virions are released from the host cell by exocytosis [26].

Importantly, the intracellular *Flavivirus* lifecycle is vulnerable to detection and inhibition by the innate immune response. As a result, DENV and ZIKV have evolved subterfuge strategies to avoid detection or prevent the actions of the innate immune response.

## 3. The Innate Immune Response to RNA Viruses

The innate immune response is the first line of defense against invading viral pathogens. This arm of the immune system generates a rapid, non-specific response aiming to control infection. The innate immune response also plays a crucial role in establishing adaptive immune responses, leading to pathogen specific and long-lasting immunological memory [31]. In general, the innate immune response is initiated by recognition of microbial components or pathogen associated molecular patterns (PAMPs) that accumulate during infection. PAMPs bind to host germline encoded pattern recognition receptors (PRRs) on the cell surface, within endosomes or in the cytoplasm, leading to recognition leads to activation of complex signaling pathways and the upregulation of multiple innate immune effector molecules and cytokines. These act directly or indirectly to control infection and promote inflammation in a temporally controlled manner. The most important class of cytokine involved in the innate response against viral pathogens are the interferons (IFN). These cytokines are responsible for orchestrating an antiviral state within infected cells, in neighboring cells and in directing immune cell activation or trafficking to control viral infection.

### 3.1. Recognition of DENV and ZIKV by the Innate Immune System

Single-stranded or double-stranded RNAs (ssRNA or dsRNA) produced as by-products of viral genome replication are commonly recognized PAMPs during *Flavivirus* infections. For DENV and ZIKV these RNA PAMPs are mainly recognized by members of the DExD/H box RNA helicase family of RIG-I like receptors (RLRs) located in the cytoplasm [32]. Detection of viral RNAs can also occur in the endosomal compartment by the membrane associated Toll-like receptors-3 or 7 (TLR) [31]. Additionally, both ZIKV and DENV infection results in host mitochondrial DNA (mitoDNA) release that is sensed by cyclic GMP-AMP synthase (cGAS) and signals through the ER associated intermediate stimulator of interferon genes (STING) [31,33,34]. Each of these pathways culminates in the phosphorylation and subsequent activation of the signaling intermediaries TANK-binding kinase 1 (TBK1) and IκB kinase-ε (IKKε). Following their activation, TBK1 and IKKε phosphorylate IFN-regulatory factors-3/7 (IRF) [31]. Additionally, IKKε phosphorylates the inhibitory subunit of nuclear factor-κB (NF-κB), leading to inhibitor degradation and subsequent activation of NF-κB [35]. Activated IRF3 and NF-κB translocate to the nucleus where they act as transcription factors to promote expression mainly of type-I IFNβ and a small subset of antiviral or proinflammatory genes [36]. Figure 3 summarizes these pathways that are discussed in more depth below.

### 3.2. RIG-I Like Receptors

The main drivers of the innate immune response against flaviviruses are the ubiquitously expressed RLRs, retinoic acid-induced gene I (RIG-I) and melanoma differentiation-associated gene 5 (MDA5) (Figure 3). RIG-I recognizes short 5′-triphosphorylated ssRNA and short dsRNAs whereas MDA5 is implicated in recognition of longer dsRNA products [9,37,38]. In structure, these PRRs contain two N-terminal caspase recruitment and activation domains (CARDs) followed by an RNA helicase domain [9]. PAMP binding causes a conformational change in the receptor that exposes the CARD interaction domains and facilitates interactions with translocation mediators such as TRIM25 and members of the 14-3-3 family of proteins [35]. The RLR translocase complex is then shuttled to the mitochondrial associated membranes where the exposed CARD domains interact with the complimentary CARD domain of signaling intermediate mitochondrial antiviral signaling protein (MAVS) [35,39]. This interaction triggers MAVS activation leading to subsequent activation of cytosolic kinases TBK1 and IKKε [31]. Interestingly, CRISPR/Cas9-mediated knockout of RIG-I but not MDA5 led to significantly increased ZIKV replication in A549 cells compared to control, indicating that RIG-I is the main sensor of ZIKV infection in these cells [40]. Furthermore, knockout of the RLR signaling intermediate MAVS was shown to enhance ZIKV infection in human placental trophoblast cell lines [41]. Similarly, siRNA mediated knockdown of RIG-I and MDA5 in the Huh7 cell line rendered them highly susceptible to DENV infection [42]. Collectively these studies demonstrate the importance of the RLRs in detection of both ZIKV and DENV infection.

### 3.3. Toll-Like Receptors

In addition to cytosolic PRR activation, viral RNAs can accumulate in the endosomal compartment where they are recognized by toll-like receptor 3 or 7 (TLR3/7) (Figure 3) [43,44]. TLRs are transmembrane glycoproteins containing an N-terminal ligand binding ectodomain, a single transmembrane domain and a C-terminal toll/interleukin-1 receptor (TIR) homology domain located in the cytoplasm [44]. Ligand binding causes receptor dimerization and the recruitment of signal transducers to the TIR domain [45]. TLR3 is more widely expressed and recognizes dsRNA whereas TLR7 is mainly expressed in plasmacytoid dendritic cells (DCs) and is activated by ssRNA [9,46]. After ligand binding and receptor dimerization the cytoplasmic TIR domain of TLR3 interacts with the TIR domain containing adapter inducing interferon-β protein (TRIF). Next, TRIF interacts with TNF receptor associated factors (TRAF3/6) and with receptor interacting proteins (RIP1/3). Comparatively, TLR7 activates a MYD88 dependent pathway that culminates in TRAF6 signaling and subsequent TBK1 and IKKε activation [9,44]. Evidence that TLR3 plays a significant role in ZIKV recognition is given by siRNA mediated gene silencing in HFF1 (foreskin fibroblast) cells rendering these cells more permissive to ZIKV [21]. Interestingly, mice deficient in TLR7 (TLR7^-/-^) supported equivalent levels of ZIKV replication compared to wildtype mice [47]. However, in this study ZIKV replication was significantly enhanced in TLR7^-/-^/MAVS^-/-^ double-knockout mice compared to wildtype and MAVS^-/-^ single-knockout mice, indicating a degree of functional redundancy between the TLR and RLR pathways. Additionally, HEK 293 cells exogenously expressing either TLR3 or TLR7 demonstrated elevated levels of downstream cytokine release following DENV infection [43].

### 3.4. cGAS-STING

Other than RNA PAMP activation of the innate immune response, DENV and ZIKV infection also results in the release of mitochondrial DNA (mitoDNA) into the cytoplasm of infected cells (33, 34). The presence of mitochondrial DNA in the cytoplasm is a hallmark of cellular damage where it acts as a potent stimulator of apoptosis and innate immune responses (Figure 3) [48]. Release of mitoDNA during DENV infection results from disruption of normal mitochondrial function. This is likely mediated by the DENV M protein that disrupts mitochondrial membrane potential [49], while the DENV NS2B/3 protease also cleaves important mitofusion proteins (MFN1 and MFN2) that are important for mitochondrial membrane fusion and homeostasis [50]. ZIKV is also known to impair mitochondrial function [51], but the molecular mechanisms that drive the release of mitoDNA during ZIKV infection are unknown. However, it is possible that these mechanisms are similar to those of DENV. Once in the cytoplasm, mitoDNA can bind directly to the cytosolic DNA sensor cGAS, resulting in a conformational change allowing cGAS to catalyze the conversion of GTP and ATP to produce the second messenger cyclic GMP–AMP (cGAMP). Next, cGAMP binds to STING located in the ER membrane causing oligomerization and translocation to the Golgi, where it activates TBK1 and IKKε [48]. Importantly knockout of cGAS in PMBCs renders these cells more susceptible to ZIKV infection and limits IFNβ production in infected cells [34]. Moreover, various human cell lines lacking STING demonstrate enhanced DENV replication in vitro [52].

Each of the above-mentioned pathways stimulates the transcription and translation of type-I IFN by infected cells. In turn the IFNs are secreted from the cell to orchestrate and amplify a broader antiviral response both within infected cells and in neighboring cells and tissues.

### 3.5. Type-I Interferons

In humans, the type-I IFNs are encoded by a cluster of related genes located on chromosome 9 [53]. They include 14 subtypes of IFNα and a single gene encoding each of IFNβ, IFNε, IFNκ, and IFNω. IFNα and IFNβ are the main IFNs produced downstream of PRR activation. In general, IFNα expression is dependent on the activation of IRFs (especially IRF7) whereas efficient IFNβ expression requires both IRF3/7 and NF-κB activation. The nuances of IRF-dependent type-I IFN expression regulation have been extensively reviewed elsewhere [54]. Collectively, the type-I IFNs are defined by their ability to signal through the type-I IFN receptor that is composed of two heterodimeric subunits (IFNAR1 and IFNAR2) and is almost ubiquitously expressed throughout the body [55].

### 3.6. Signalling from the Type-I IFN Receptor

Once produced in response to detection viral infection, secreted type-I IFN binds to the extracellular domains of its cognate receptor (IFNAR1/2) expressed on the same cell or on neighboring cells (Figure 4). Importantly, most cell types respond to type-I IFNs due to the almost ubiquitous expression of IFNAR1/2 [55,56]. Ligand binding causes hetero-dimerization of the receptor subunits bringing the intracellular domains of the receptor into close proximity [57]. Each of the IFNAR1 and IFNAR2 intracellular domains are pre-associated with tyrosine kinases that are activated upon receptor dimerization by close proximity trans-phosphorylation [53]. IFNAR1 is associated with tyrosine kinase 2 (TYK2) and IFNAR2 is associated with janus kinase 1 (JAK1) respectively [58,59]. Once activated JAK1 and TYK2 phosphorylate the intracellular domains of the IFNAR subunits, allowing docking of signal transducer and activators (STAT1 and STAT2). Once docked to the receptor, JAK1 and TYK2 phosphorylate STAT1 and STAT2 at tyrosine residues Y701 or Y690 respectively [31]. Phosphorylation leads to the formation of STAT1/2 heterodimers, nuclear translocation and subsequent complexing with IRF9 [56]. This hetero-trimeric complex called interferon stimulated gene factor 3 (ISGF3) then binds to IFN-stimulated response elements (ISRE) in the proximal promoter regions of over 100 ISGs, up-regulating their transcription and translation [11]. ISGs encode various proteins that carry out a range of effector or regulatory functions [31]. Effector functions are varied and include inhibition of viral entry, inhibition of protein synthesis, alterations to cellular metabolism, degradation of viral proteins or genetic material and inhibition of viral egress [11]. Regulatory ISGs include PRRs and signaling partners of these pathways as well as immunomodulatory molecules and negative regulators responsible for controlling immune cell activation and trafficking or returning the cell to homeostasis, respectively [60].

### 3.7. Inhibition of ZIKV and DENV Infection by Interferon Stimulated Genes

Type-I IFNs are important in controlling *Flavivirus* infection through the production of ISGs. These ISGs play a wide variety of roles in the innate antiviral response and have been extensively reviewed elsewhere [11]. Importantly, some ISGs can act directly to inhibit various stages of the ZIKV and DENV virus lifecycle. Due to the shared aspects of their lifecycles some of these ISGs are similarly effective against both viruses. Several of these direct acting ISGs are known and their mechanisms of action are described below.

Two complimentary studies have demonstrated that Interferon Stimulated Gene 15 (ISG15) protects against ZIKV infection [61,62]. Mature ISG15 is a 15 kDa member of the ubiquitin family of proteins that plays various roles in the innate immune response that have been extensively reviewed [63]. One of the main functions of ISG15 is the covalent modification of target proteins via ISG15 conjugation (ISGylation) disrupting target protein localization and protein-protein interactions. Additionally, ISG15 has immune modulatory functions through non-covalent protein interactions and by acting as an immune cell signaling molecule. One study found that ZIKV infected ISG15^-/-^ mice had increased severity of retinal lesions and impaired antiviral responses compared to wildtype mice and this led to lower expression of other ISGs like RIG-I and IFI6 [62]. A follow-up study by the same group extended these observations to show that ZIKV infection in human primary corneal epithelial cells (HCEC) induced expression of ISG15 RNA and protein, and that siRNA mediated knockdown of this expression lead to increased ZIKV infection in these cells [62]. Conversely, heterologous expression of ISG15 protein was able to ameliorate this effect. Furthermore, this group found that ISG15 expression was important for both reducing ZIKV entry into host cells, and for inhibiting viral replication once inside the cell. Similarly, ISG15 inhibits DENV infection. DENV infection has been shown to upregulate expression of ISG15 in RAW264.7 cells, and its silencing increased DENV replication in these cells [64]. Furthermore, infection with DENV increased the total amount of ISGylated proteins in the cell, suggesting a link between the ISG conjugation activity of ISG15 with the observed antiviral effect [64].

Members of the Interferon-Inducible Transmembrane (IFITM) protein family also protect against both ZIKV and DENV infection. As their name suggests these proteins are found inserted into cellular membranes, most commonly localizing in late endosomes, and can interfere with fusion of viral and host membranes following viral entry [65]. Importantly, siRNA mediated knockdown of IFITM1 or 3 was shown to increase ZIKV infection in human cell lines [66]. This effect could be rescued with overexpressed protein but relied on the protein’s endosomal localization. Consistent with their localization to endosomal membranes, knockdown of IFITM1 or 3 was shown to impact very early steps in the viral replication cycle following entry [66]. Likewise ectopically expressed IFITM2 or IFITM3 have been shown to reduce DENV infection in human cell lines to similar levels to those observed when treating cells with 100 U/mL IFNα [67]. Like ZIKV, this inhibitory effect was observed to impact the DENV lifecycle at a step prior to the initial round of viral RNA translation.

Another ISG shown to directly inhibit ZIKV and DENV infection is interferon alpha-inducible protein 6 (IFI6). IFI6 is a 13 kDa protein that is known to be involved in cell survival and counteracting viral-mediated apoptosis [68]. Increased expression of IFI6 was shown to reduce ZIKV replication and prevented ZIKV mediated cell death in the Huh7 (liver origin) cell line [69]. In this study, IFI6 localized to the ER near ZIKV RCs suggesting that it may play a role in inhibiting viral replication or virion production. Furthermore, this antiviral effect of IFI6 was independent of ZIKV protein stability or polyprotein processing. Likewise Huh7.5 cells stably transduced with a lentivirus IFI6 expression vector demonstrated decreased DENV replication compared to an empty vector control [70]. This study also demonstrated that CRISPR-mediated knockout of endogenous IFI6 expression increased DENV replication in infected cells.

Furthermore, the virus inhibitory endoplasmic reticulum associated interferon inducible protein (Viperin) has been shown in multiple studies to reduce ZIKV and DENV replication [71,72,73]. Viperin is a 42 kDa protein and as the name suggests is normally associated with ER membranes. Importantly, Viperin has antiviral effects against a wide range of viruses in both RNA and DNA families [74,75,76]. The first study to investigate the importance of Viperin in ZIKV infection found that Viperin was induced in response to ZIKV infection and overexpressed Viperin restricted ZIKV replication in human cell lines [71]. Additionally, murine embryonic fibroblasts (MEFs) derived from Viperin knockout mice were more permissive to ZIKV replication compared to wildtype MEFs. Furthermore, the anti-ZIKV action of Viperin relied on the highly conserved C-terminal end of the protein [74]. A second study confirmed Viperin’s antiviral effect, finding that Viperin interacted directly with the ZIKV NS3 protein resulting in its degradation and reduced viral replication [72]. Similarly, DENV inhibition by Viperin is also dependent on the C-terminal region of the protein and its interaction with the DENV NS3 protein [73].

Aside from these ISGs that are known to inhibit both ZIKV and DENV infection, other direct acting ISGs have been independently validated for either virus. Future studies may prove these ISGs are effective against both viruses, or they may reveal virus-specific activity. One of these ISGs is interferon-inducible factor 16 (IFI16). This ISG has multiple roles in modulating expression of viral proteins and activating the STING pathway during infection [77]. Overexpressed IFI16 was shown to reduce infection of both the +ssRNA alphavirus chickungunya virus (CHIKV) and ZIKV in human skin fibroblasts [78]. No specific mechanism for this effect was investigated. To our knowledge, IFI16 has not been independently validated for antiviral activity against DENV infection. However, the role of IFI16 in promoting STING activation suggests that this is a strong possibility.

Other ISGs known to inhibit DENV infection are the ArfGAP with dual pleckstrin homology (PH) domains 2 (ADAP2) protein and the tripartite motif 69 protein (TRIM69). ADAP2 is most highly expressed in the heart, and skeletal muscle and is known to regulate the ADP ribosylation factor (Arf) family of proteins via its GTPase activating protein (GAP) function [79]. Arf proteins are involved in regulating vesicular trafficking and cytoskeletal organization. Importantly, ectopically expressed ADAP2 has been demonstrated to restrict DENV infection by inhibiting GAP mediated trafficking of incoming DENV containing vesicles [80]. TRIM69 mediates protein ubiquitination through its E2 conjugation enzymatic activity. It localizes to the cytoplasm and inside the nucleus of cells [81]. TRIM69 has been shown to directly interact with DENV NS3, resulting in its polyubiquitination and subsequent degradation to inhibit DENV replication [82]. To our knowledge, the roles of both ADAP2 and TRIM69 in protection against ZIKV infection have not been investigated.

In the context of natural infection, multiple ISGs are expressed in concert and as a result target many of the stages of DENV and ZIKV replication simultaneously. The combined effects of ISGs to inhibit ZIKV and DENV infection has applied strong selective pressure on these viruses to evolve mechanisms to evade detection by the cell or prevent the production of ISGs through blocking the IFN signaling pathway.

## 4. Common Strategies Employed by Flaviviruses to Evade the IFN Response

As obligate intracellular pathogens, flaviviruses have evolved many ways to avoid detection by host intracellular PRRs. For example, the 7-methylguanylate cap that is incorporated on the 5′ end of the +ssRNA genome by the NS5 MTase domain mimics the appearance of host mRNA (18). 5′ capping assists host translational initiation factors to bind the viral RNA and prevents RNA degradation by endonucleases in the cytoplasm [83]. Importantly, 5′ capping also interferes with the recognition of vRNA as ‘non-self’ by the MDA5 PRR, limiting the production of IFN by infected cells [84]. In addition to hiding from detection, capping also avoids the antiviral effects of the interferon induced protein with tetratricopeptide repeats (IFIT) protein family that can bind to and sequester vRNA of uncapped or cap mutant *Flavivirus* genomes [85].

The second mechanism commonly used by flaviviruses to avoid detection is the induction of membrane rearrangements allowing for the formation of the RC as part of the virus lifecycle [86]. As mentioned previously, for flaviviruses these structures are formed from modified ER membranes by the coordinated action of multiple host and viral proteins. Replication organelles form as a series of membranes that surround the dsRNA replication intermediate, acting to physically segregate this potent viral PAMP from detection by cytoplasmic PRRs like RIG-I [87,88]. In addition, these replication organelles limit the antiviral activity of the ISG encoded MXA protein, likely by blocking MXA-mediated recognition of the forming viral nucleocapsid [89]. Evidence in support of this theory is provided in one study comparing DENV to JEV replication where they observed a greater degree of dsRNA in the cytosol during JEV infection compared to DENV and this correlated to increased IFN production in JEV infected cells [90]. Ultrastructural analysis of the ZIKV RC shows it to be highly similar to that of DENV, suggesting their related function [91].

Aside from genome capping and the formation of replication organelles, several flaviviruses evade IFN production by expression of subgenomic *Flavivirus* RNA (sfRNA) [92]. *Flavivirus* sfRNAs are formed from the incomplete degradation of the viral genome by cellular 5′ to 3′ exonucleases [93]. Specifically, conserved stem-loop or dumbbell RNA structures within the 3′UTR stall exonuclease activity and result in the production of short RNA sequences [94]. Importantly sfRNAs generated during ZIKV infection antagonize the activity of both RIG-I and MDA5 [95], although a full mechanism of action has not been elucidated for this interaction. Likewise, DENV sfRNA inhibits RIG-I mediated IFN production. This effect is governed by a sequence-specific interaction between DENV sfRNA and the tripartite motif containing 25 protein (TRIM25) [96]. TRIM25 functions as an RNA binding protein and a ubiquitin ligase, responsible for the polyubiquitination of activated RIG-I leading to sustained signal transduction [35].

### 4.1. ZIKV-Specific Mechanisms to Evade IFN Responses

In addition to strategies common to all flaviviruses, some evasion mechanisms have been characterized for ZIKV that may be unique for this virus. These mechanisms prevent ISG production by interfering with PRR-mediated IFN production, or by directly targeting signaling intermediaries downstream of the IFNAR1/2 receptor.

One mechanism that limits the production of IFN is mediated by ZIKV NS4A that inhibits RIG-I and MDA5 signaling and has been demonstrated independently by two groups [41,97]. Collectively these studies found that ectopically expressed NS4A binds directly to the N-terminal CARD domain MAVS. This binding competitively inhibited MAVS interaction with activated RIG-I or MDA5, leading to potent inhibition of downstream type-I IFN production.

ZIKV can also prevent the translocation of activated RIG-I and MDA5 to the mitochondrial membranes by acting on members of the 14-3-3 protein family. These proteins (14-3-3ε and 14-3-3η) act as mitochondrial targeting chaperones that are required for translocation of RIG-I and MDA5 respectively, facilitating their interaction with MAVS [98,99]. Importantly, overexpressed ZIKV NS3 protein in HEK 293T cells was able to competitively bind to both 14-3-3ε and 14-3-3η via a conserved binding motif (64-RLDP-67) [100]. This sequence was found to contain a central negatively charged Aspartic acid residue (D66) that acted as a phospho-mimetic to compete with RIG-I or MDA5 binding. Mutation of this binging motif within the full length ZIKV genome attenuated viral replication compared to wildtype virus in A549 cells [100].

ZIKV can also inhibit the cGAS-STING pathway via the actions of NS1. One study found that ectopically expressed ZIKV NS1 interacts directly with host de-ubiquitinase USP8 to facilitate the deubiquitination of caspase-1, increasing its stability [34]. In turn caspase-1 proteolytically cleaves cGAS, reducing the production of IFN in the cell. Additionally, the ZIKV protease NS2B/3 mediates STING cleavage. Using exogenous expression in HEK 293T cells it was shown that ZIKV NS2B/3 cleaved human but not mouse STING [101]. This study extended these observations to natural ZIKV infection in human fibroblasts by detection of STING cleavage products during infection. The reduction of cGAMP mediated STING activation as a result of ZIKV infection inhibited the production of IFN by infected cells.

Other NS proteins also contribute to limit IFN production downstream of MAVS, TLR and cGAS-STING pathways by targeting the shared signaling intermediaries TBK1, IKKε, or IRF3. Ectopically expressed NS1 and NS4B interact directly with TBK1, preventing TBK1 oligomerization and phosphorylation mediated activation [102]. Interestingly, another study found that NS1 mediated TBK1 inhibition was specific to recent outbreak strains that had evolutionarily acquired a 188-Val substitution mutation [103]. Additionally, overexpressed ZIKV NS5 in HEK 293 cells was demonstrated to directly interact with IKKε [104]. This direct interaction resulted in reduced IKKε protein levels and phosphorylation, thereby preventing the activation of IRF3. In another study, NS5 was also shown to inhibit IRF3 activation by direct binding to endogenous IRF3 in studies involving co-immunoprecipitation of overexpressed NS5 protein [103].

Downstream of the IFN receptor, ZIKV also acts to suppress JAK-STAT signal transduction. The best characterized of these mechanisms is the ZIKV NS5 mediated degradation of STAT2 protein. ZIKV NS5 can bind to STAT2 and initiate its degradation in a proteasome dependent manner [105]. Interestingly, this occurs with human but not mouse STAT2 protein, partially explaining the difference in species adaptation of ZIKV and difficulties associated with infecting IFN competent mice [106]. A separate study found that overexpression of NS5 leading to STAT2 degradation also resulted in reduced STAT1 phosphorylation in cells overexpressing NS5 [107]. One study also demonstrated that ZIKV NS2B/3 expression induced the degradation of JAK1 protein in a proteasome-dependent manner leading to a reduction in IFN mediated ISG expression [102]. In addition to the roles of NS proteins in inhibition of IFNAR signaling, ZIKV binding to the attachment factor Axl on the cell surface also inhibits IFN signaling. In a study using microglial cell lines, ZIKV binding was shown to activate the C-terminal kinase domain of Axl that in turn acted to negatively regulate the type-I signaling pathway via induction of SOCS1 protein expression [108].

### 4.2. DENV-Specific Mechanisms to Evade the IFN Response

Some of the mechanisms that govern DENV-mediated IFN evasion closely reflect those of ZIKV, demonstrating their close evolutionary relationship. These tend to be evasion strategies mediated by the more conserved viral proteins, such as the NS3 helicase/protease and NS5 RdRp/MTase. However, there are also mechanisms that differ entirely in their action and are unique to DENV.

One mechanism preventing the production of IFN by DENV is mediated by NS2B. Overexpressed DENV NS2B directly interacts with cGAS and causes its degradation by auto-phagolysosomes, reducing STING-mediated IFN production [33].

Additionally, DENV NS2B/3 proteolytically cleaves human but not mouse STING in a similar manner to ZIKV [52]. This species-specific cleavage was dependent on the presence of an NS3 cleavage site in human STING. Mutation of this cleavage site was able to restore DENV-mediated induction of IFNβ. DENV NS3 also contributes to evasion of IFN by non-proteolytic actions. HEK 293T cells expressing DENV NS3 demonstrated impaired RIG-I translocation to MAVS in response to Sendai Virus infection. Like ZIKV, this interaction was also dependent on inhibition of RIG-I binding to 14-3-3ε via a conserved phospho-mimetic binding motif at the same location within NS3 (64-RxEP-67) [109]. However, the charged residue mimicking the natural phosphorylation site was found to be a Glutamic acid (Glu66) rather than Aspartic acid residue as was found for the ZIKV NS3 protein. Furthermore, DENV NS2A and NS4B from multiple DENV serotypes inhibit PRR mediated IFN production via targeting TBK1 and IRF3. Ectopically expressed DENV NS2A and NS4B were shown to specifically inhibit TBK1 auto-phosphorylation, and reduced total IRF3 protein levels [110]. Moreover, this same study found that NS4A from serotype-1 was in addition uniquely able to contribute to TBK1 inhibition [110]. This additional evasion mechanism may contribute to the enhanced virulence of DENV1. Overexpressed DENV NS2B/3 in HEK 293/TLR3 expressing cells also directly interacts with the N-terminal Kinase domain of IKKε, inhibiting kinase activity and reducing IFN production [111].

Downstream of IFN receptor activation DENV NS4B blocks STAT1 phosphorylation and nuclear translocation [112]. DENV NS2A and NS4A were also shown to inhibit ISRE promoter activity downstream of IFNβ signaling in HEK 293T cells [112]. For NS4B this effect was later found to depend on the N-terminal signal peptide of the NS4B protein and was enhanced by natural cleavage between the NS4A-2K-NS4B fragment [113]. For NS2A and NS4A the mechanism driving their IFN evasion properties has not yet been elucidated. Finally, similar to ZIKV, DENV NS5 mediates human STAT2 degradation in a proteasome dependent manner [114]. This similarity is highlighted by a recent study that found the specific interacting residues of NS5 and its binding mode with human STAT2 were highly conserved between ZIKV and DENV [115]. However, in contrast to ZIKV this requires natural viral processing of NS5 from the polyprotein and is dependent on the ubiquitin ligase UBR4 [116].

Importantly, this intricate web of viral:host molecular interactions for both ZIKV and DENV means the innate immune system competes with the virus in a race to establish an antiviral state or a state of immune-suppressed infection. The outcome of this race largely determines the outcome of natural infection or responses to vaccination. A summary of ZIKV and DENV mediated IFN-evasion mechanisms is given in Table 1 below.

## 5. Exploitation of Enhanced Type-I IFN Responses for Effective Vaccine Development

Aside from directly controlling viral replication, the type-I IFNs also have a significant role in enhancing adaptive immune responses to viral pathogens. Expression of the type-I receptor on both T and B cells is required for efficient activation of antibody responses in mice [117,118]. Additionally, type-I IFN enhances isotype switching via activation of antigen-presenting dendritic cells (DCs) [119]. Because several of the ISGs that are expressed in response to IFNs are chemokines and chemokine receptors, type-I IFNs also influence immune cell migration and recruitment. For example, in the presence of type-I interferon, DCs express greater levels of chemokine receptor CCR7 which is important in the generation of primary immune responses, and greater levels of CXCL10 that are required for recruitment of Th1 memory lymphocytes [120]. Furthermore, type-I IFN influences cytotoxic T cell expansion and memory formation [121]. Our understanding of the effect that type-I IFN has on immune responses to *Flavivirus* infection have vastly improved due to the development of immune-competent small animal models. While beyond the scope of this review, small animal models of DENV and ZIKV infection are of vital importance in pre-clinical testing of the efficacy and safety of vaccine candidates. Recent advances in this area have been comprehensively reviewed elsewhere [106,122,123].

Collectively, the combined action of type-I IFNs are important to promote efficient immune activation and therefore IFN or IFN-stimulating adjuvants are often used to enhance vaccine responses. For example, co-administrated type-I IFN can act as an adjuvant to improve vaccine responses against influenza in mice [124]. Furthermore, the cationic polysaccharide chitosan is an adjuvant currently used in vaccines that results in improved Th1 responses compared to other adjuvants like alum salts that promote mainly Th2 responses [125]. It was recently discovered that chitosan mediated this action by increasing type-I IFN production, influencing DC maturation and leading to improved antigen specific Th1 responses following vaccination [126]. Therefore, enhancing type-I IFN responses in vaccination settings is a proven way to enhance vaccine efficacy, especially where Th1 responses are important for effective control against viral pathogens [127].

Aside from its potential use as an adjuvant for vaccines, type-I IFN can also be induced naturally by live-attenuated vaccines leading to long lasting immunity. One example of this is the YF-17D vaccine licensed for YFV. One study investigating the transcriptomic profile of humans immunized with YF-17D found that the type-I IFN response was the most highly activated immune signaling pathway [128]. The type-I IFN response was induced early and transiently, returning to baseline by day 14. Importantly, the YF-17D vaccine is known to offer highly effective, lifelong protection in most patients [129].

Mutation of the genetic regions involved in viral evasion of the IFN response has shown promise in developing new live-attenuated vaccine candidates. One study performing functional profiling on the Influenza virus genome found a series of mutations within NS1 that conferred IFN hypersensitivity of these viruses [130]. When these IFN sensitive mutants were introduced into a lethal model of mouse influenza, the virus generated robust type-I IFN and adaptive immune responses but was highly attenuated, resulting in a 100% survival rate of infected mice. Inoculation with these IFN sensitive mutant virus strains protected against homologous and heterologous viral challenge in the lethal mouse model. Another example of this approach to generate potential live-attenuated vaccines was recently reported for ZIKV [131]. In this study, a full-length infectious clone of ZIKV was subjected to site directed mutagenesis focused on residues in ZIKV NS4B that are important for IFN evasion, namely a C100S mutation. These mutant viruses were propagated in Vero cells that are deficient in IFN production and used to infect mice. Whereas mice infected with wildtype virus displayed 100% lethality upon challenge, the C100S mutant virus did not result in significant weight loss or death, indicating that it was successfully attenuated. During infection, the C100S mutant was found to induce stronger type-I IFN and antigen specific T cell responses compared to the parent strain. Notably, vaccination with the C100S mutant protected mice from a lethal ZIKV challenge.

These examples demonstrate the importance of type-I IFN responses in generating protective immunity in vaccines. Furthermore, they highlight how the knowledge of viral evasion strategies can be used for the targeted, rational design of new live-attenuated vaccine candidates.

## 6. Conclusions

DENV and ZIKV are significant human pathogens that lack appropriate control measures. The development of safe and effective vaccines is important in the global fight against these pathogens. The immune response to both viruses is heavily underpinned by an appropriate type-I IFN response. However, both viruses have developed ways to counteract this aspect of innate immunity to cause infection and disease in humans. Understanding the molecular interactions of these viruses with the type-I IFN response is important and may assist in improving current vaccine strategies though adjuvants or aid in producing future mutant live-attenuated vaccine candidates that are unable to evade type-I IFN responses.

## Figures and Tables

**Figure 1 vaccines-08-00530-f001:**
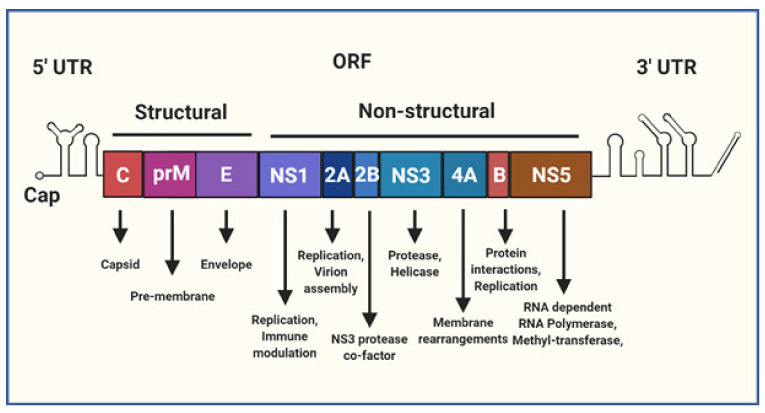
A schematic representation of the Flavivirus genome. The genome is a single positive-sense single-stranded RNA (+ssRNA) molecule that is roughly 11 kb in size, capped at the 5′ terminus and flanked by 5′ and 3′ untranslated regions (UTR). The central open reading frame (ORF) encodes a polyprotein that is cleaved into individual structural and non-structural (NS) viral proteins, the text below lists some of their known functions as reviewed in [17].

**Figure 2 vaccines-08-00530-f002:**
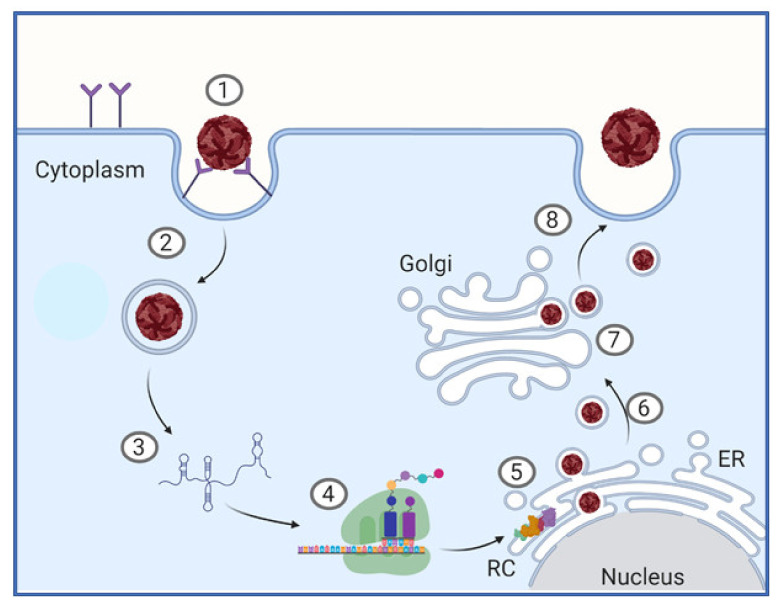
Stages of the *Flavivirus* lifecycle. (1) Attachment and receptor-mediated endocytosis. (2) Membrane fusion and particle disassembly. (3) Genome release into the cytoplasm. (4) Polyprotein translation. (5) Replication complex (RC) formation and genome replication. (6) Virion packaging. (7) Transportation through the trans-Golgi network and virion maturation. (8) Virion egress by exocytosis.

**Figure 3 vaccines-08-00530-f003:**
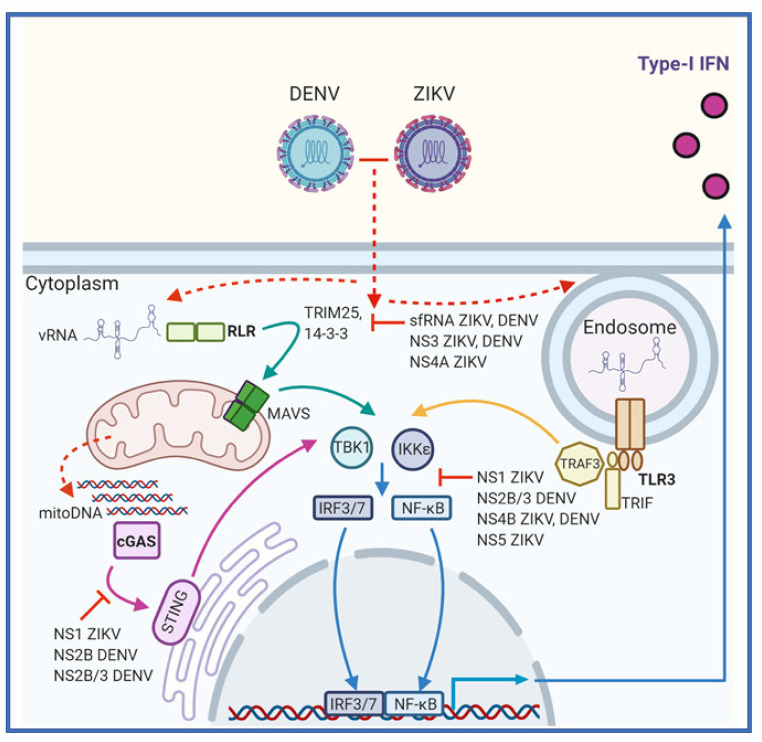
Recognition of Zika Virus (ZIKV) and Dengue Virus (DENV) by the innate immune system and viral evasion of pattern recognition receptor (PRR)-mediated interferons (IFN) production. ZIKV and DENV infections are sensed by multiple PRRs inside the cell. or pathogen associated molecular pattern (PAMP) vRNA from the replication stage of the virus lifecycle activates RLRs in the cytosol (green arrows) or Toll-like receptors (TLRs) in the endosome (gold arrows). Alternatively, both ZIKV and DENV infection results in release to mitoDNA into the cytoplasm that is sensed by cyclic GMP-AMP synthase (cGAS), activating stimulator of interferon genes (STING) (purple arrows). Each of these pathways leads to phosphorylation of kinases TBK1 and IKKɛ that activate IRF3/7 and NF-κB. These transcription factors then upregulate the production of type-I IFNβ (blue arrows). However, ZIKV and DENV have evolved evasion mechanisms (red) to prevent IFN production that target multiple stages of these pathways.

**Figure 4 vaccines-08-00530-f004:**
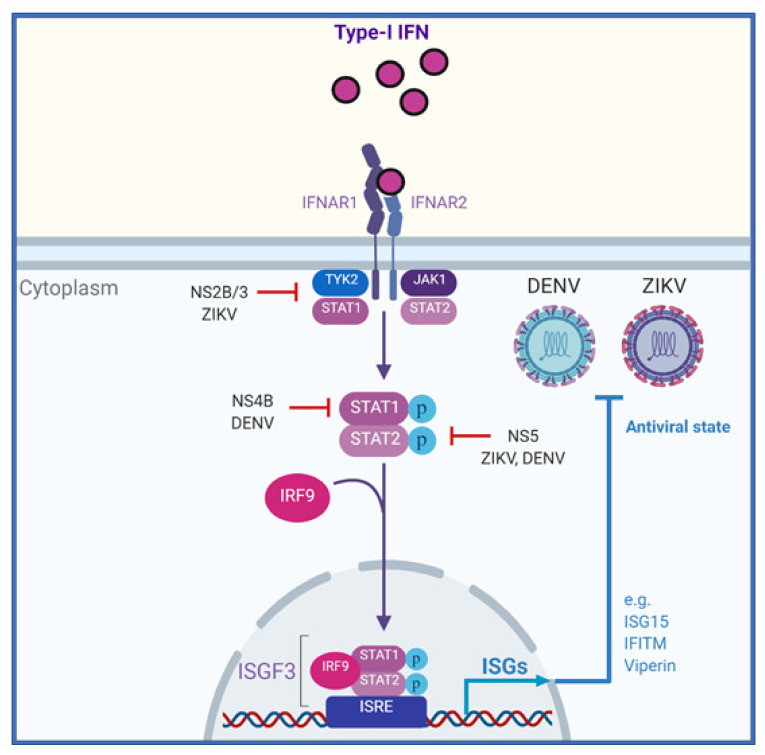
The molecular interactions of ZIKV and DENV with the type-I IFN pathway. Once produced and secreted from infected cells type-I IFNs bind to their cognate cell surface receptor, activating janus kinase-signal transducer and activator (JAK-STAT) signaling (black arrows). This leads to the phosphorylation of STAT1 and STAT2, their heterodimerzation and subsequent complexing with IRF9 to form the complex ISGF3. ISGF3 then transcriptionally upregulates interferon stimulated genes (ISGs) that have antiviral effects against ZIKV and DENV. Several stages of the IFN pathway are inhibited by ZIKV and DENV mediated evasion mechanisms (red).

**Table 1 vaccines-08-00530-t001:** Summary of virus specific IFN evasion mechanisms for both ZIKV and DENV

ZIKV-Mediated IFN Evasion Mechanisms			
Viral Factor	Immune Pathway	Host Target	References
NS4A	RLR	Binds directly to MAVS	[41,97]
NS3	RLR	Competitively binds to both 14-3-3ε and 14-3-3η	[100]
NS1	cGAS-STING	Binds USP8 leading to cGAS cleavage	[34]
NS1 and NS4B	RLR, TLR, cGAS-STING	Interacts with TBK1	[102]
NS5	RLR, TLR, cGAS-STING	Interacts with IKKɛ	[104]
NS5	RLR, TLR, cGAS-STING	Direct binding to IRF3	[103]
NS5	IFNAR1/2	STAT2 degradation	[105]
NS2B/3	IFNAR1/2	JAK1 degradation	[102]
Viral attachment	IFNAR1/2	Binding to Axl on the cell surface inducing SOCS1 expression	[108]
**DENV-Mediated IFN Evasion Mechanisms**			
**Viral Factor**	**Immune Pathway**	**Host Target**	**References**
NS2B	cGAS-STING	cGAS degradation	[33]
NS2B/3	cGAS-STING	STING cleavage	[52]
NS3	RLR	Competitively binds to both 14-3-3ε	[109]
NS2A and NS4B	RLR, TLR, cGAS-STING	TBK1 inhibition and reduced IRF3 protein levels	[110]
NS2B/3	RLR, TLR, cGAS-STING	Interacts with IKKɛ	[111]
NS4B	IFNAR1/2	Blocking STAT1 phosphorylation	[112]
NS5	IFNAR1/2	STAT2 degradation	[114]

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
