# Peer review of "The Molecular Interactions of ZIKV and DENV with the Type-I IFN Response"

_vaccines, 2020, doi:10.3390/vaccines8030530_

Round 1

Reviewer 1 Report

The manuscript from Rosa C. Coldbeck-Shackley et al. describes the type-I IFN response to flaviviruses like Zika and Dengue in much detail. The review is written highly understandable and delivers an excellent insight into the IFN-topic.

The authors should consider the following points to improve their manuscript:

Major points:

Point 1: DENV definition in lane 10, 22:  DENV and ZIKA are not two viruses. Same in the Conclusion section. All this implies that DENV is one strain. Please change your definition to the four serotypes, DENV-1-4. Please consider this also for the statement of amino acid identity in lane 70.

Point 2: For the discussion of vaccine strategies against flavivirus it seems to be important to enhance the understanding of the first human humoral response, the priming by natural infection. We can also say priming by the virus variant, genotype or serotype from that specific area. In the introduction of the manuscript you are ending your first paragraph by touching upon the Dengvaxia problem. Then immediately you start with: The type-I-IFNs…..

I think that is a hard break in your introduction and nothing is said on the Dengvaxia topic hereafter. But for flavivirus vaccine design this is a really important finding that age is not the surrogate to vaccinate people. The Dengvaxia vaccine is helpful for people that had been primed by natural infection. We now know that the primed neutralizing response is highly type-specific, not only serotype specific and therefore priming people with antigen that do not match with ME variants present in their area put them on risk. The four Dengvaxia antigens DENV-1-4 are from old DENV strains and from different countries. To get into this story, Maíra Aguiar and Nico Stollenwerk were one of the first scientist who reported on that problem (Dengvaxia: age as surrogate for serostatus, https://doi.org/10.1016/S1473-3099(17)30752-1). Therefore, Dengvaxia can be seen as a booster but not as a vaccine for priming. This kind of prime and boost effect was observed in the old monkey vaccine studies where monkey were challenged with two different DENV serotypes.

Thus, in general, every vaccine strategy in the DENV and ZIKV field will be confronted with naïve or naturally (multiple) primed individuals, and the antigen variation seen in different areas. The major reason is that the prM Ab response is group- and the neutralizing response is highly type-specific. There are also reports on DENV induced ZIKV ADE Abs.

I would recommend considering this in your manuscript to make it more interesting.

Point 3: Another aspect is the 6mA modification of the flavivirus RNA genome that has an impact on flavivirus replication and therefore its live cycle. Non-modified genomes are packed into virus particles and modified genomes stay in the cell for translation. Please see. N6-Methyladenosine in Flaviviridae Viral RNA Genomes Regulates Infection, https://doi.org/10.1016/j.chom.2016.09.015.  

And in the next reference it was shown that ISGs have an impact on m6A modified RNA from HBV: Interferon-stimulated gene 20 (ISG20) selectively degrades N6-methyladenosine modified Hepatitis B Virus transcripts. PLoS Pathog . 2020 Feb 14;16(2):e1008338. doi: 10.1371/journal.ppat.1008338. eCollection 2020 Feb.

I would recommend considering such ISG effects in your manuscript as well.

Minor points:

Immunologists like abbreviations. When you read chapter 4 and find PAMP at lane 374 you have to search the manuscript to find its explanation that is on lane 135. Same for  ISG lane 395, you have to go to lane 49. Other abbreviations for stem-loop or dumbbell are never used again (lane 384). I therefore suggest summarizing them all in a table of a new chapter #8.  That would be very helpful for the newbie.

Lane 15: I believe that the flaviviruses do not use a myriad number of escape mechanisms. Especially in the abstract I recommend not using such a statement. You actually enumerate them in your review.

Lane 119-120: This statement implies that the immune system is seeing only the mature ME complex. This is not the case since about 50% of the DENV surface is covered by non-infectious, uncleaved prME dimers. Please see also point 7 in your Fig 2. This is very important since the uncleaved part of prM is the major target for the antibody dependent enhancement (ADE) antibodies and that’s also the reason why the Dengvaxia vaccine is not safe in flavivirus seronegative individuals. The UK group has done a lot on human mAbs to prM to show their ADE specificity.

Reviewer 2 Report

Coldbeck-Shackley et al. review the current knowledge on the interaction between Zika virus (ZIKV) and Dengue virus (DENV) with the type-I interferon (IFN) response. The topic is timely and relevant. The manuscript is well organized and carefully written and the figures and tables are appropiate.

Major points

  1. Only section 6 is focused to the relation between interferon and vaccines. Authors should include an additional specific section about animal models of ZIKV and DENV infection and IFN response. This would be of special interest in order to choose the right model for vaccine testing.

Minor

  1. Flaviviruses and flavivirus in italics. In my opinion flaviviruses should not be in italics when commonly used. Only when the taxonomic group Flavivirus is referered and it should be also capitalized.
  2. Line 70. Flavivirus is a genus, not a family (Flaviviridae)
  3. Fig. 3. The flavivirus particles seems to be inside a vesicle at the ER and when trafficking through the Golgi complex. This representation could be cofusing.
  4. The structures of human STAT2 in complex with the NS5 of ZIKV and DENV has been recently solve, This information could be added to the review (Nat Struct Mol Biol. 2020 Aug 10. doi:10.1038/s41594-020-0472-y).
